# Effects of Tofacitinib Therapy on Circulating Tumour-Associated Antigens and Their Relationship with Clinical, Laboratory and Vascular Parameters in Rheumatoid Arthritis

**DOI:** 10.3390/biom15050648

**Published:** 2025-04-30

**Authors:** Enikő Sebestyén, Dóra Csige, Péter Antal-Szalmás, Ágnes Horváth, Edit Végh, Boglárka Soós, Zsófia Pethő, Nóra Bodnár, Attila Hamar, Levente Bodoki, Dorottya Kacsándi, Róza Földesi, Edit Kalina, Gábor Nagy, György Kerekes, Béla Nagy, Katalin Hodosi, Szilvia Szamosi, Péter Árkosy, Gabriella Szűcs, Zoltán Szekanecz, Éva Szekanecz

**Affiliations:** 1Department of Oncology, Faculty of Medicine, University of Debrecen, 4032 Debrecen, Hungary; senci0827@gmail.com (E.S.); arkosy.peter@med.unideb.hu (P.Á.); szevadr17@gmail.com (É.S.); 2Department of Rheumatology, Faculty of Medicine, University of Debrecen, 4032 Debrecen, Hungary; dora.csige96@gmail.com (D.C.); kis.horvathagi@gmail.com (Á.H.); veghe22@gmail.com (E.V.); soosbogi@gmail.com (B.S.); pethozsofi0331@gmail.com (Z.P.); drbodnarnora@gmail.com (N.B.); attilahamar.2010@gmail.com (A.H.); bodoki.levente@gmail.com (L.B.); kacsandi.dorottya@gmail.com (D.K.); khodosi@gmail.com (K.H.); szamosi.szilvi@gmail.com (S.S.); szucs.gabriella@med.unideb.hu (G.S.); 3Department of Laboratory Medicine, Faculty of Medicine, University of Debrecen, 4032 Debrecen, Hungary; antalszp@med.unideb.hu (P.A.-S.); rfoldesi@med.unideb.hu (R.F.); ekalina@med.unideb.hu (E.K.); nagyb80@gmail.com (B.N.J.); 4Intensive Care Unit, Department of Medicine, Faculty of Medicine, University of Debrecen, 4032 Debrecen, Hungary; gkerekesg@gmail.com

**Keywords:** rheumatoid arthritis, tumour-associated antigens, tofacitinib, vascular pathophysiology

## Abstract

Introduction: Tumour-associated antigens (TAA) have been implicated in cell adhesion and cancer metastasis formation, but also in inflammatory processes, such as rheumatoid arthritis (RA). There has been little information about the possible associations of TAAs with RA-related clinical and laboratory parameters, with impaired vascular pathophysiology in RA, as well as about the effects of antirheumatic drugs on TAA production. Therefore, we determined the effects of one-year tofacitinib treatment on TAA levels, as well as correlations of TAA levels with various RA-associated and vascular parameters. Patients and methods: Altogether, 26 RA patients received 5 mg bid or 10 mg bid tofacitinib treatment for 12 months. Ultrasound-based functional vascular assessments, such as common carotid intima-media thickness (ccIMT), brachial artery flow-mediated vasodilation (FMD) and carotid-femoral pulse-wave velocity (cfPWV), were determined at various timepoints. Serum concentrations of TAAs, including carcinoembryonic antigen (CEA), CA15-3, CA19-9, CA125, CA72-4, human epididymis protein 4 (HE4) and tissue polypeptide antigen (TPA), as well as various cytokines (TNF-α, IL-6, IL-8, VEGF) and PECAM-1 were determined by flow cytometry using a bead-based multiplex assay (LEGENDplex). Results: As previously determined and published, one-year tofacitinib treatment effectively suppressed disease activity and inflammation. Serum CA15-3 and HE4 levels significantly decreased both after 6 and 12 months compared to baseline (*p* < 0.05). CA19-9 levels significantly increased both after 6 and 12 months, while CEA levels transiently increased after 6 months versus baseline (*p* < 0.05). CA125, CA72-4 and TPA levels did not change over time. In various regression analyses, TAA levels showed variable, significant, positive associations with the 28-joint disease activity score (DAS28), CRP, ESR, RF, IL-6, TNF-α, IL-8 and PECAM-1 (*p* < 0.05). In addition, TAAs variably correlated with ccIMT and cfPWV (*p* < 0.05). Moreover, one-year changes in TAA levels variably correlated with DAS28, RF and some cytokines (*p* < 0.05), as well as with changes in DAS28, HAQ, CRP, ESR, IL-6, VEGF and ccIMT from baseline to 12 months (*p* < 0.05). Conclusions: JAK inhibition might decrease the levels of some TAAs and increase those of others. TAA levels might be associated with RA-related and vascular biomarkers. These results suggest that TAAs might play a role in inflammatory processes and vascular pathology underlying RA.

## 1. Introduction

Janus kinases (JAKs) have been involved in the signalling of multiple cytokines [1,2], as well as in synovial inflammation underlying rheumatoid arthritis (RA) [3,4]. Four JAK inhibitors, including tofacitinib, a pan-JAK inhibitor of the JAK1 and JAK3 isoforms, have been approved for the treatment of RA [5]. JAK inhibitors, together with biologic disease-modifying agents, have long been used, either alone or in combination with methotrexate, for the treatment of RA [5].

RA has been associated with increased cardiovascular (CV) morbidity and mortality [6,7,8]. Abnormal CV pathophysiology occurs very early, even preclinically in RA patients [6,7,9,10]. Ultrasound-based imaging techniques are suitable to detect vascular pathophysiology [11]. Endothelial dysfunction of the brachial artery, common carotid atherosclerosis and increased arterial stiffness are indicated by impaired endothelium-dependent, flow-mediated vasodilation (FMD), increased common carotid intima-media thickness (ccIMT) and carotid-femoral pulse-wave velocity (cfPWV), respectively [6,11]. Anti-inflammatory therapies including JAK inhibitors might dampen arterial inflammation and improve impaired vascular pathophysiology in RA [12,13,14,15].

Tumour-associated antigens (TAA) may, apart from cancer cells, become expressed on the surface of inflammatory cells. Some of them serve as cell adhesion molecules (CAM). These TAAs might be involved in the perpetuation of inflammation in RA [16,17] and other inflammatory rheumatic diseases [17,18]. These TAAs may be shed from the cell surface and can be detected in the sera of RA as well as cancer patients [16,17]. Members of the carcinoembryonic antigen (CEA) family, also known as CD66a-e molecules, belong to the immunoglobulin superfamily of CAMs. These antigens contain carbohydrate motifs, including Lewis-x (Le-x) and sialyl-Lewis-x (sLe-x), and they bind to E-selectin [19,20,21]. We and others detected CEA antigens in the RA synovial tissue [19,21]. Moreover, higher CEA levels were more frequently found in RA [16,22], and CEA has been associated with RA-related interstitial lung disease (RA-ILD) [23,24]. Apart from the CEA family, other TAAs may also be associated with inflammatory diseases. Among these TAAs, CA15-3 is an epitope of MUC1 and has been associated primarily with breast cancer [16,17]. CA15-3 as well as KL-6 are alternative epitopes of MUC1 [25,26], and both have been associated with RA-ILD [25,26,27,28,29]. Regarding the RA joint, MUC1 was expressed in both the synovial lining and sublining layer within the RA synovial tissue. MUC1 expression in RA was higher than in controls and correlated with joint destruction scores. Blockade of MUC1 by siRNA inhibited the synovial expression of proinflammatory cytokines and the invasion of synoviocytes [30]. The immune determinant of CA19-9 is the carbohydrate antigen sialyl-Lewis a (sLe(a)), a ligand for E-selectin [16,17,31]. CA19-9 levels are elevated in various gastrointestinal malignancies [16]. Cancer cell sLe(a)–E-selectin interactions regulate metastasis formation [32]. Both E-selectin and various sialyl-Lewis antigens have been implicated in the pathogenesis of RA synovitis [33,34]. In some studies, elevated or more frequently higher circulating CA19-9 levels were found in the sera of RA patients [16,22,35]. CA19-9 has also been associated with RA-ILD [23,28,29,36]. CA125 is also known as MUC16, another mucin. CA125 is mainly associated with ovarian cancer and its progression [16]. CA125 has been proposed to link RA disease activity with the development of ovarian cancer [37]. Moreover, CA125 has also been associated with RA-ILD [28,29,36]. CA125 and rheumatoid factor (RF) levels correlated with each other in RA [38]. CA72-4 is a non-specific marker of various malignancies, including gastrointestinal, ovarian, breast and lung cancer [16,39]. There has been almost no information on the possible role of CA72-4 in RA or RA-ILD. On the other hand, CA72-4 levels might rather be elevated in gout [40]. Human epididymis protein 4 (HE4) is a marker of epithelial ovarian cancer, lung cancer and some other malignancies [41]. Among non-malignant diseases, we and others have associated HE4 with various lung diseases, including RA-ILD, cystic fibrosis, Sjögren’s syndrome and others [23,42,43,44,45]. In RA-ILD, HE4 correlates with disease severity and is a marker of poor prognosis [45]. Finally, tissue polypeptide antigen (TPA) is a mixture of cytokeratin 8, 18 and 19. TPA is a specific marker of bladder cancer but, in combination with other TAAs, is an indicator of proliferation in most solid malignancies, such as lung cancer [46,47]. There have been no reports on the relationship between RA and TPA. In our previous study, TPA levels were not elevated in RA.

Some years ago, we were among the first to measure TAA production in RA. We assessed circulating CEA, CA19-9, CA125, CA15-3 and CA72-4 levels in RA patients and controls. There were significantly more RA patients showing abnormally high levels of CA125, CA19-9 and CA15-3 compared to controls. In addition, RA patients had elevated mean serum levels of CA125 and CA19-9 versus controls. In RA, CEA highly correlated with RF [16].

As described above, some TAAs have been associated with RA-ILD. However, little information has become available regarding the connection of TAAs with other aspects of RA or the effects of antirheumatic therapies on TAA production. Therefore, in the present study, we assessed seven TAAs in tofacitinib-treated RA patients for the first time to study the effects of JAK inhibition on TAA production. Furthermore, we correlated TAA production with RA disease activity, inflammatory markers, cytokines, angiogenic growth factors, an adhesion molecule (PECAM-1) and vascular pathophysiology as determined by ccIMT, FMD and cfPWV.

## 2. Patients and Methods

### 2.1. Patients and Study Design

Thirty patients with active RA were consecutively recruited for this interventional study. Patient characteristics are presented in Table 1. Inclusion criteria included definitive diagnosis of RA according to the 2010 European League Against Rheumatism (EULAR)/American College of Rheumatology (ACR) classification criteria for RA [48]; moderate-high 28-joint disease activity score (DAS28 > 3.2) at baseline and clinical indication of targeted therapy. Patients were either naïve to any targeted therapies (*n* = 16) or initiated tofacitinib after stopping a biologic followed by an appropriate washout period (*n* = 14). Exclusion criteria included inflammatory diseases other than RA, acute/recent infection, standard contraindications to JAK inhibition, uncontrolled CV disease or hypertension, chronic renal or liver failure and malignancy within 10 years.

The 30 enrolled patients received either 5 mg or 10 mg tofacitinib twice daily (bid) treatment arms. In the EU, the 5 mg bid dose has been approved for the treatment of RA; however, in many countries, the 10 mg bid dose is also in use. The dose was randomly assigned to each patient in a 1:1 ratio. All patients received tofacitinib in combination with either methotrexate (MTX) (*n* = 23) or leflunomide (*n* = 7). MTX and leflunomide had been taken in stable doses at least 1 year prior to the present study. No dose changes of these DMARDs were allowed throughout the course of the study. Although most patients may have received corticosteroids prior to the study, none of the patients had been on corticosteroids for at least 3 months prior to and during the study.

Clinical assessments were performed at baseline and after 6 and 12 months of therapy. Four patients (2 on each arm) completed the six-month follow-up but did not complete the one-year treatment. Twenty-six patients completed the one-year treatment period and were included in the data analysis.

The study was approved by the Hungarian Scientific Research Council Ethical Committee (approval No. 56953-0/2015-EKL, approval date: 8 March 2016). Written informed consent was obtained from each patient, and assessments were carried out according to the Declaration of Helsinki and its amendments.

### 2.2. Clinical Assessment

First, a detailed medical history was taken. We inquired about the history of cardiovascular disease as well as current smoking, experience of chest pain resembling angina pectoris, hypertension and diabetes mellitus during the last 2 years before the start of this study by a questionnaire (Table 1). Further clinical assessments, including physical examination, were performed at baseline, and after 3, 6 and 12 months of tofacitinib therapy.

### 2.3. Assessment of Vascular Physiology by Ultrasound

Ultrasound-based functional vascular assessments, such as ccIMT, FMD and cfPWV measurements, were carried out at baseline and during the follow-up after 6 and 12 months. Details of investigations were thoroughly described and published previously [12]. In the present study, FMD, ccIMT and cfPWV data are only used in the correlation analysis.

### 2.4. Laboratory Measurements and Disease Activity

Erythrocyte sedimentation rate (ESR) was determined by a standard procedure. Serum high sensitivity C-reactive protein (hsCRP; normal: ≤5 mg/L) and IgM rheumatoid factor (RF; normal: ≤50 IU/mL) were measured by immunoturbidimetry (Cobas^®^ c502/702, Roche Diagnostics, Basel, Switzerland). ACPA (CCP) autoantibodies were detected in serum samples using a second-generation Immunoscan-RA CCP2 ELISA test (Euro Diagnostica, Malmö, Sweden; normal: ≤25 IU/mL). The assays were performed according to the manufacturer’s instructions.

Disease activity of RA was calculated as DAS28-CRP (3 variables).

### 2.5. Tumour-Associated Antigens and Other Biomarkers

Serum concentrations of TAAs, including CEA (normal: <3.4 μg/L), CA15-3 (normal: <25 kIU/L), CA19-9 (normal: <34 kIU/L), CA125 (normal: <35 kIU/L), CA72-4 (normal: <6.9 kIU/L), HE4 (normal: <70 pmol/L) and TPA (normal: <75 U/L), as well as those of tumour necrosis factor α (TNF-α; normal <136.3 pg/mL), interleukin 6 (IL-6; normal <3.7 pg/mL), IL-8/CXCL8 (normal: <120.1 pg/mL), vascular endothelial growth factor (VEGF; normal <204.2 pg/mL) and platelet-endothelial cell adhesion molecule 1 (PECAM-1; normal <26.9 ng/mL) were determined by flow cytometry using a bead-based multiplex assay (Human Angiogenesis Panel 1, 10-plex, LEGENDplex, BioLegend, San Diego, CA, USA) and analysed by LEGENDplex software (https://www.biolegend.com/en-ie/immunoassays/legendplex/support/software, accessed on 28 April 2025). Normal values were determined by validation using normal human serum samples. The assays were performed according to the manufacturer’s instructions. The effects of tofacitinib treatment on circulating cytokine and angiogenic growth factor levels were published separately [49]. In the present study, these data are only used in the correlation analysis.

### 2.6. Statistical Analysis

Statistical analysis was performed using SPSS version 26.0 (IBM, Armonk, NY, USA) software. Data are expressed as the mean ± SD for continuous variables and percentages for categorical variables. The distribution of continuous variables was evaluated by Kolmogorov–Smirnov test. Continuous variables were assessed by paired two-tailed *t*-test and Wilcoxon test. Nominal variables were compared between groups using the chi-squared or Fisher’s exact test, as appropriate. Correlations were determined by Pearson’s analysis. Univariable and multivariable regression analysis using the stepwise method were applied to investigate independent associations between TAA levels (dependent variables) and all other parameters (independent variables). The β standardized linear coefficients showing linear correlations between two parameters were determined. The B (+95% CI) regression coefficient indicated independent associations between dependent and independent variables during changes.

General linear model (GLM) repeated measures analysis of variance (RM-ANOVA) was performed to determine the additional effects of various biomarkers on 12-month changes in TAA levels (dependent variable). Two-way RM-ANOVA analysis was also conducted to determine the correlations between one-year changes (changes between baseline and 12 months) in TAA levels and one-year changes in other parameters. In the RM-ANOVA and two-way RM-ANOVA analyses, partial η^2^ is given as indicator of effect size, with values of 0.01 suggesting minor, 0.06 medium and 0.14 major effects [49,50]. *p* values < 0.05 were considered significant.

The reliability of the vascular ultrasound measurements was tested by inter-item correlation and two-way, mixed, single rater intraclass correlation (ICC) before [12,13].

## 3. Results

### 3.1. Characteristics of Patients

These data have been published before based on other studies emerging from the very same cohort [12,49]. There were two patients in the 5 mg bid and two patients in the 10 mg bid group who dropped out. Thus, 26 patients completed the study [12,49]. The characteristics of these 26 patients are included in Table 1.

### 3.2. Effects of Tofacitinib Therapy on Disease Characteristics and Vascular Pathophysiology

As published before, one-year tofacitinib treatment was highly effective in controlling RA. Tofacitinib treatment significantly improved DAS28, CRP and HAQ both after 6 and 12 months [12,49]. Regarding vascular pathophysiology, in brief, FMD and cfPWV showed no changes, while ccIMT increased over time [12]. In this study, we only used the clinical, disease activity and vascular ultrasonography data to correlate them with the TAA results. Thus, none of the data to be presented below have been published yet.

### 3.3. Effects of JAK Inhibition on TAA Levels

In this RA cohort, serum CA15-3 levels significantly decreased both after 6 months (18.4 ± 5.5 kIU/L; *p* = 0.049) and 12 months of tofacitinib therapy (18.3 ± 6.3 kIU/L; *p* = 0.031) in comparison to baseline (19.4 ± 6.1 kIU/L) (Figure 1B). Similarly, HE4 levels were also significantly reduced after 6 months (57.6 ± 22.9 pmol/L; *p* = 0.001) and 12 months (61.1 ± 24.5 pmol/L; *p* = 0.014) versus baseline (68.7 ± 26.6 pmol/L) (Figure 1F).

CA19-9 levels significantly increased both after 6 months (11.1 ± 8.4 kIU/L; *p* = 0.014) and 12 months (11.4 ± 9.2 kIU/L; *p* = 0.008) in comparison to baseline (9.1 ± 5.8 kIU/L) (Figure 1C). CEA levels were only transiently elevated after 6 months (3.46 ± 2.21 μg/L; *p* = 0.029) compared to baseline (2.82 ± 2.06 μg/L), but this increase was not significant after 12 months (3.05 ± 1.97 μg/L; *p* = 0.124) (Figure 1A).

We did not observe any significant changes in CA125 (12.1 ± 3.9 kIU/L; 11.5 ± 3.7 kIU/L; 11.7 ± 4.7 kIU/L; Figure 1D), CA72-4 (2.30 ± 3.46 kIU/L; 1.90 ± 2.42 kIU/L; 2.15 ± 2.60 kIU/L; Figure 1E) and TPA levels (37.7 ± 24.0 U/L; 31.6 ± 16.4 U/L; 38.2 ± 33.2 U/L; Figure 1G) between baseline, 6 months and 12 months, respectively.

We also counted the number of patients with TAA levels higher than the normal upper limit at different time points (Table 2). The number of patients with elevated serum CA15-3 levels clearly decreased after 6 months (*n* = 3) and 12 months (*n* = 3) compared to baseline (*n* = 7). On the other hand, the number of patients with CEA levels above the normal upper limit clearly increased after 6 months (*n* = 13) and 12 months (*n* = 11) compared to baseline (*n* = 8). There were no observed changes in this respect regarding CA19-9, CA125, CA72-4, HE4 and TPA (Table 2).

### 3.4. Correlations of TAA Levels with Other Parameters in RA

In the univariable regression analysis, where TAAs were the dependent variables, CA15-3 levels showed variable, statistically significant, positive associations with DAS28, IL-6, and PECAM-1 (*p* < 0.05), while HE4 levels were associated with RF, TNF-α, and IL-8 (Table 3). The other TAAs showed more limited correlations. CEA was associated with TNF-α and RF, CA19-9 only with RF, CA125 with PECAM-1, CA72-4 with DAS28 and IL-6, and TPA with ESR, RF and HAQ (*p* < 0.05) (Table 3). The multivariable regression analysis confirmed associations between CA15-3 and IL-6 and PECAM-1, CA72-4 and IL-6, HE4 and TNF-α and IL-8, as well as TPA and ESR (*p* < 0.05) (Table 3).

We assessed whether TAAs might determine vascular pathophysiology over time. Indeed, in the univariable regression analysis, CEA, CA15-3, CA125 and HE4 variably correlated with ccIMT, while CEA, CA125 and TPA were variably associated with cfPWV (*p* < 0.05) (Table 3). The multivariable regression analysis confirmed the associations of CA15-3 and HE4 with ccIMT and that of TPA with cfPWV (*p* < 0.05) (Table 3).

GLM RM-ANOVA analysis was conducted to assess determinants of TAA changes, certainly also connected to tofacitinib treatment, between baseline, 6 and 12 months. The one-year change in CEA levels was determined by disease duration (*p* = 0.037). Moreover, the 12-month change in CA19-9 levels was determined by baseline RF and TNF-α levels (*p* < 0.05). Changes in CA72-4 levels over time were determined by baseline DAS28, IL-6 and PECAM-1 (*p* < 0.05) (Table 4).

In the two-way RM-ANOVA analysis, we found variable correlations between TAA level changes and changes in other parameters over time. As indicated in Table 4, one-year changes in CEA, CA15-3, CA19-9, CA72-4 and HE4 levels variably correlated with changes in DAS28, HAQ, CRP, ESR, IL-6 and VEGF over time (*p* < 0.05).

Finally, in the RM-ANOVA analysis, baseline CA125 levels were associated with one-year changes in ccIMT (*p* = 0.046) (Table 4).

To better understand key correlations, we show them in Table 5.

## 4. Discussion

To the best of our knowledge, this is the first study on the effects of a JAK inhibitor on seven TAAs. These TAAs have been implicated in metastasis formation but, in some cases, also in inflammatory processes [16,17,18,23,29]. On the other hand, little information has become available on the possible involvement of TAAs in RA [16,33]. CEA and MUC1 are expressed in the RA synovium [19,21,30]. The production of some TAAs, such as CEA, CA15-3, CA19-9 and CA125, might be increased in RA [16,22,35]. Moreover, elevated serum levels of some TAAs have been associated with RA-ILD [23,24,25,26,28,29,36,43,44,45]. Yet, there have been no studies on the associations of these TAAs with other RA parameters and on the possible effects of antirheumatic therapies on these TAAs.

In the present study, tofacitinib significantly decreased CA15-3 and HE4 levels after 6 months and 12 months of treatment. In our previous study, significantly more RA patients had abnormally high CA15-3 levels than controls [16]. Moreover, twice as many RA patients had CA15-3 levels above the upper limit at baseline than after 6 and 12 months. We have no further information on the role of CA15-3 and HE4 in RA, only in RA-ILD [25,26,27,28,29,43,44]. Yet, CA15-3 is an epitope of MUC1, and MUC1 shows elevated expression in the RA synovial tissue compared to controls. Moreover, MUC1 was associated with joint destruction [30]. Therefore, CA15-3, most probably as an epitope of MUC1, might indeed play an important role in RA synovitis. We have no information on the possible involvement of HE4 in RA synovitis. In our hands, one-year JAK inhibition suppressed disease activity and inflammation in RA, and this was accompanied by a decrease in CA15-3 and HE4 levels over time.

Tofacitinib therapy resulted in a transient increase in CEA levels after 6 months, which then decreased again after 12 months. Moreover, more patients had abnormally high CEA levels after 6 and 12 months than at baseline. JAK inhibition also increased CA19-9 levels over the one-year period; however, there were absolutely no patients with abnormally high CA19-9 levels at any time point. It is not fully clear why tofacitinib therapy increased the serum levels of these two TAAs. We had found high CEA expression in the RA synovial tissue before [33]. It is possible that tofacitinib might resulted in a clearance of CEA from the synovium, and this was reflected by increased circulating CEA.

The suppressive effects of tofacitinib on CA15-3 and HE4 and, at the same time, the stimulating effects of this drug on CEA and CA19-9 levels need further studies and explanation. All these TAAs have been involved in inflammation, including cell adhesion. It is possible that they act differently in a given inflammatory environment, involve different signalling pathways, and therefore, they might be differentially influenced by JAK inhibition [16,23]. Yet, none of the patients had CA19-9 levels above the normal upper limit at any time point, indicating that CA19-9 might be less important in this respect than CA15-3 or HE4. Tofacitinib therapy had no effects on CA125, CA72-4 and TPA levels between baseline and 12 months.

We performed correlation analysis on TAAs and other RA-related parameters. Even if, as discussed above, the levels of some TAAs increased, others decreased, and others did not change upon tofacitinib treatment, all investigated TAAs showed correlations with various clinical, inflammatory and vascular biomarkers at different time points during treatment. In general, CA15-3 and HE4 showed numerous correlations with disease activity, markers of systemic inflammation, RF, cytokines and PECAM-1. CEA, CA19-9 and TPA also correlated with RF, as well as cytokines and PECAM-1. In previous studies, CEA and CA125 correlated with RF in RA [16,38]. As there have been no other similar correlation analyses, we could not further compare our data with those from other investigators. Thus, various TAAs at different time points might reflect disease activity, inflammation and the production of various pro-inflammatory mediators. Similarly, changes in TAA levels might reflect changes in systemic inflammation and disease activity caused by JAK inhibition. Concerning vascular pathophysiology, some TAAs, including CEA, CA125, HE4, and TPA, showed positive correlations with ccIMT and cfPWV. This suggests that some TAAs, which might also serve as cell adhesion molecules, can be involved in vascular pathology underlying RA. There have been absolutely no reports on the possible involvement of these TAAs in cardiovascular disease, only, as discussed above, in ILD [25,26,27,28,29,43,44]. Finally, the GLM RM-ANOVA and two-way RM-ANOVA analyses confirmed that changes in various TAA levels over time might be determined by changes in disease activity, CRP, ESR, cytokines and PECAM-1, together with the tofacitinib therapy between baseline and 12 months. These correlations also suggest that changes in TAA levels might reflect tofacitinib-induced changes in inflammation and disease activity. In the absence of similar studies, our results could not be compared with other reports. In conclusion, TAAs, primarily CA15-3 and HE4 but also others, might show two-way associations with other parameters of RA including DAS28, markers of inflammation (CRP, ESR), cytokines (TNF-α, IL-6, IL-8, VEGF), PECAM-1, as well as the ccIMT and cfPWV vascular pathophysiology markers.

This study suggests the role of some TAAs in RA and in RA-associated vascular pathology. The possible role of these TAAs in synovial and vascular pathophysiology remains somewhat unknown. As discussed above, most TAAs exert adhesive properties. They are released from damaged epithelial and other cells. TAAs might be implicated in various inflammatory and fibrotic mechanisms. Therefore, they might be involved in cell adhesive processes underlying joint inflammation and vascular pathology. JAK inhibition might alter TAA release by dampening inflammation and suppressing disease activity. Yet, it is difficult to explain why tofacitinib lowered the serum levels of some TAAs and increased those of others. It is possible that various TAAs might act differently in inflammatory states and upon immunosuppression [22,23,42]. Also, the clinical utility of these TAAs needs further confirmation in larger studies. TAAs might have clinical implications, as TAAs are correlated with disease activity, markers of inflammation, and several pro-inflammatory mediators, as well as ccIMT and cfPWV. Thus, TAAs could be used as inflammatory markers, as well as for risk stratification and monitoring vascular risk. In conclusion, TAAs might reflect the ongoing inflammatory processes, link synovial inflammation and vascular pathology and thus, in addition to ILD, might be used as biomarkers during the follow-up of the patients.

Our study might have some strengths and limitations. The major strengths include the originality, meaning that TAAs have been assessed in context with various pathological parameters and vascular markers. Also, TAA data have been prospectively obtained before and after treatment. One of the strengths of the present study is that this might be the very first one on the effects of a JAK inhibitor on TAA levels and the associations of TAA with various RA parameters and vascular pathophysiology markers. Limitations might include the lack of control groups, the use of other csDMARDs in combination, as well as the relatively low number of patients. One major limitation of this study is the lack of functional mechanistic studies to explain differential regulation of TAAs. In addition, although some studies indicate that TAAs might be associated with RA-ILD [25,26,27,28,29,43,44], due to the lack of available data, we were unable to assess the RA-ILD issue in our cohort.

## 5. Conclusions

JAK inhibition might decrease the levels of some TAAs and increase those of others. CA15-3 and HE4 stand out in terms of clearly decreasing levels over time, as well as the great number of correlations with markers of RA and RA-related vascular pathology. Other markers, including CA19-9, CA125, CA72-4 and TPA, showed some associations with RA-related and vascular biomarkers. The effects of JAK inhibition on TAA levels and the correlations of TAAs with various parameters might reflect the possible involvement of TAAs in the pathogenesis of RA and RA-related vascular disease. Moreover, some TAAs might serve as therapeutic targets in RA. Certainly, further studies are warranted in order to more clearly delineate the role of TAAs in RA and its comorbidities.

## Figures and Tables

**Figure 1 biomolecules-15-00648-f001:**
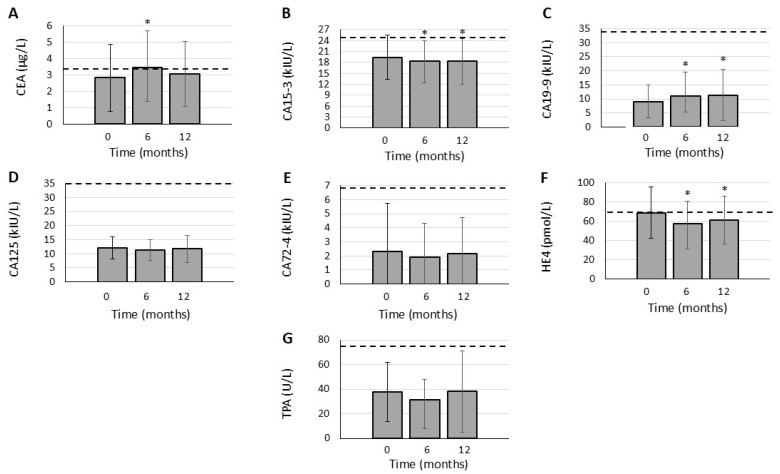
One-year changes in (**A**) CEA, (**B**) CA15-3, (**C**) CA19-9, (**D**) CA125, (**E**) CA72-4, (**F**) HE4 and (**G**) TPA levels upon one-year tofacitinib treatment. *N* = 26 in all cases. Dashed lines show normal upper limits. “*” indicates significant differences compared to baseline (*p* < 0.05).

**Table 1 biomolecules-15-00648-t001:** Patient characteristics.

	Tofacitinib-Treated Patients
Number of patients (*n*)	26
Female:male ratio	23:3
Age (years), mean ± SD (range)	51.9 ± 9.7 (27–69)
BMI (kg/m^2^), mean ± SD (range)	30.3 ± 7.4 (20.8–51.4)
Positive CV history, *n* (%)	6 (23.1)
Positive history of hypertension, *n* (%)	13 (50.0)
Positive history of diabetes mellitus, *n* (%)	2 (7.7)
Smoking (current), *n* (%)	7 (26.9)
Disease duration (years), mean ± SD (range)	7.5 ± 4.8 (1–21)
RF positivity, *n* (%)	22 (84.6)
Anti-CCP positivity, *n* (%)	22 (84.6)
DAS28 (baseline), mean ± SD	5.12 ± 0.82

Abbreviations: BMI, body mass index; CCP, anti-cyclic citrullinated peptide; CV, cardiovascular; DAS28, 28-joint disease activity score; RF, rheumatoid factor; SD, standard deviation.

**Table 2 biomolecules-15-00648-t002:** Number of RA patients with TAA levels above the normal upper limit at different time points.

TAA	Baseline	6 Months	12 Months
CEA	8	13	11
CA15-3	7	3	3
CA19-9	0	0	0
CA125	0	0	0
CA72-4	2	1	1
HE4	7	5	7
TPA	2	1	2

**Table 3 biomolecules-15-00648-t003:** Univariable and multivariable analysis of TAA associations with other parameters.

Dependent Variable	Independent Variable	Univariable Regression Analysis				Multivariable Regression Analysis			
		β	*p*	B	95% CI	β	*p*	B	95% CI
TAAs as Dependent Variables									
CEA-0	TNF-0	0.487	0.012	0.057	0.014–0.100				
CEA-6	RF-0	0.391	0.048	0.004	0–0.008				
	IMT-6	0.520	0.006	10.384	3.197–17.570	0.520	0.006	10.384	3.197–17.570
CA153-0	DAS28-0	0.442	0.023	4.911	1.222–7.888				
	IL6-0	0.565	0.003	0.075	0.029–0.121	0.565	0.003	0.075	0.029–0.121
	PECAM1-0	0.486	0.012	0					
CA153-6	DAS28-6	0.530	0.005	4.584	1.493–7.676				
	IL6-0	0.631	0.001	0.075	0.036–0.113				
	PECAM1-6	0.593	0.001	0		0.593	0.001	0	
CA153-12	DAS28-6	0.484	0.012	4.826	1.146–8.503				
	IL6-0	0.655	<0.001	0.089	0.046–0.133				
	PECAM1-12	0.647	<0.001	0	0–0.001	0.647	<0.001	0	0–0.001
CA199-12	RF-0	0.418	0.034	0.017	0.001–0.033				
CA125-6	IMT-6	0.466	0.017	15.477	3.082–27.871				
CA125-12	PECAM1-12	0.463	0.017	0					
	IMT-12	0.414	0.035	17.753	1.330–34.177	0.442	0.001	18.937	4.874–32.999
CA724-0	DAS28-0	0.425	0.030	1.937	0.199–3.675				
	IL6-0	0.449	0.021	0.034	0.005–0.062	0.430	0.041	0.032	0.006–0.059
HE4-0	RF-0	0.403	0.041	0.049	0.002–0.096				
	TNF-0	0.489	0.011	0.743	0.185–1.302	0.480	0.006	0.730	0.228–1.232
	IL8-0	0.429	0.029	0.392	0.044–0.739	0.419	0.015	0.383	0.081–0.684
HE4-6	IL8-0	0.426	0.030	0.334	0.035–0.634				
	IMT-0	0.530	0.005	109.8	35.7–183.8	0.530	0.005	109.8	35.7–183.8
	IMT-6	0.412	0.037	79.973	5.353–154.6				
HE4-12	RF-0	0.463	0.017	0.052	0.010–0.093				
	RF-6	0.456	0.019	0.061	0.011–0.112				
	RF-12	0.402	0.042	0.056	0.002–0.110				
	IL8-0	0.541	0.004	0.453	0.156–0.750	0.441	0.006	0.370	0.116–0.624
	IMT-0	0.583	0.002	129.054	53.35–204.7	0.494	0.003	109.391	42.389–176.4
	IMT-12	0.452	0.020	76.543	12.948–140.1				
TPA-6	IMT-6	0.441	0.037	60.853	4.016–117.69				
TPA-12	HAQ-12	0.401	0.042	19.747	0.752–38.741				
	ESR-0	0.394	0.047	0.613	0.010–1.215				
	ESR-12	0.501	0.009	0.918	0.250–1.587	0.435	0.010	0.797	0.207–1.387
	RF-0	0.464	0.017	0.700	0.014–0.127				
	PWV-12	0.423	0.031	8.560	0.843–16.278				
TAAs as Independent Variables									
IMT-6	HE4-6	0.112	0.037	0.002	0–0.004				
IMT-12	CEA-6	0.430	0.028	0.028	0.003–0.053				
	CA153-6	0.430	0.028	0.011	0.001–0.021	0.433	0.013	0.011	0.003–0.020
	HE4-6	0.461	0.018	0.003	0.001–0.005	0.464	0.009	0.003	0.001–0.005
	HE4-12	0.452	0.020	0.003	0–0.005				
PWV-12	CEA-6	0.405	0.040	0.301	0.015–0.587				
	TPA-6	0.486	0.012	0.049	0.012–0.086	0.486	0.012	0.049	0.012–0.086
	TPA-12	0.423	0.031	0.009	0.002–0.040				

Abbreviations: CI, confidence interval; DAS28, 28-joint disease activity score; ESR, erythrocyte sedimentation rate; IL, interleukin; IMT, carotid intima-media thickness; PECAM, platelet-endothelial adhesion molecule; PWV, pulse-wave velocity; RF, rheumatoid factor; TNF, tumour-necrosis factor.

**Table 4 biomolecules-15-00648-t004:** GLM RM-ANOVA and two-way RM-ANOVA analysis of associations between TAA changes and changes in other biomarkers between baseline and 12 months. (A) RM-ANOVA. (B) Two-way RM-ANOVA.

(A)				
Dependent Variable	Effect	F	*p*	Partial η^2^
TAAs as Dependent Variables				
CEA (0-6-12)	disease duration	4.188	0.037	0.149
CA19-9 (0-6-12)	RF-0	3.426	0.041	0.125
	TNF-0	12.414	<0.001	0.519
CA72-4 (0-6-12)	DAS28-0	6.392	0.012	0.210
	IL6-0	14.273	0.001	0.373
	PECAM1-0	9.119	<0.001	0.275
TAAs as Independent Variables				
IMT (0-6-12)	CA125-0	3.520	0.046	0.234
(**B**)				
**Dependent Variable**	**Effect**	**F**	** *p* **	**Partial** **η^2^**
TAAs as Dependent Variables				
CEA (0-6-12)	DAS28 (0-6-12)	39.124	<0.001	0.765
	HAQ (0-6-12)	5.568	0.010	0.317
	CRP (0-6-12)	9.240	0.001	0.435
	ESR (0-6-12)	12.539	<0.001	0.511
	IL-6 (0-6-12)	4.351	0.024	0.266
	VEGF (0-6-12)	6.350	0.006	0.346
CA15-3 (0-6-12)	CRP (0-6-12)	8.348	0.002	0.410
	ESR (0-6-12)	10.364	0.001	0.463
	IL-6 (0-6-12)	3.794	0.037	0.240
	VEGF (0-6-12)	6.165	0.007	0.399
CA19-9 (0-6-12)	DAS28 (0-6-12)	10.029	0.001	0.455
	HAQ (0-6-12)	4.532	0.021	0.274
	CRP (0-6-12)	8.348	0.002	0.410
	ESR (0-6-12)	11.804	<0.001	0.496
	IL-6 (0-6-12)	5.091	0.014	0.298
	VEGF (0-6-12)	6.645	0.005	0.356
CA72-4 (0-6-12)	DAS28 (0-6-12)	4.122	0.029	0.256
	CRP (0-6-12)	6.885	0.004	0.365
	ESR (0-6-12)	10.561	0.001	0.468
	IL-6 (0-6-12)	4.606	0.020	0.277
	VEGF (0-6-12)	6.155	0.007	0.399
HE4 (0-6-12)	DAS28 (0-6-12)	5.882	0.008	0.329
	HAQ (0-6-12)	7.288	0.003	0.378
	VEGF (0-6-12)	4.511	0.022	0.273
	IMT (0-6-12)	7.586	0.003	0.378

The numbers 0, 6 and 12 represent baseline, 6-month and 12-month results. Abbreviations: CRP, C-reactive protein; DAS28, 28-joint disease activity score; ESR, erythrocyte sedimentation rate; HAQ, health assessment questionnaire; IMT, carotid intima-media thickness; IL, interleukin; VEGF, vascular endothelial growth factor.

**Table 5 biomolecules-15-00648-t005:** Summary of key correlations between TAAs and other parameters at various time points.

TAA	DD	DAS28	CRP	ESR	HAQ	RF	TNF-α	IL-6	IL-8	VEGF	PECAM-1	IMT	PWV
CEA	+	+	+	+	+	+	+	+		+		+	+
CA15-3		+	+	+				+		+	+	+	
CA19-9		+	+	+	+	+	+	+		+			
CA125											+	+	
CA72-4		+	+	+				+		+	+		
HE4		+			+	+	+		+	+		+	
TPA				+	+	+						+	+

“+” represents positive correlations. Abbreviations: CRP, C-reactive protein; DAS28, 28-joint disease activity score; DD, disease duration; ESR, erythrocyte sedimentation rate; HAQ, health assessment questionnaire; IL, interleukin; IMT, carotid intima-media thickness; PECAM, platelet-endothelial cell adhesion molecule; PWV, pulse-wave velocity; RF, rheumatoid factor; TNF, tumour necrosis factor; VEGF, vascular endothelial growth factor.

## Data Availability

The raw data supporting the conclusions of this article will be made available by the authors on request.

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
