# Peer review of "Effects of Tofacitinib Therapy on Circulating Tumour-Associated Antigens and Their Relationship with Clinical, Laboratory and Vascular Parameters in Rheumatoid Arthritis"

_biomolecules, 2025, doi:10.3390/biom15050648_

Round 1
Reviewer 1 Report
Comments and Suggestions for Authors
This study is novel in its investigation of tumor-associated antigens (TAAs) in the context of rheumatoid arthritis (RA) and their modulation by JAK inhibition (tofacitinib). It offers potentially valuable insights into RA pathophysiology and vascular comorbidities. This study is the first to correlate TAAs with vascular parameters (ccIMT, cfPWV) in RA. Also, this study provides longitudinal data with robust biomarker analysis, and addresses both inflammatory and vascular components of RA.
However, several issues need to be addressed.
- I think the major limitation of this study is no functional mechanistic studies to explain differential regulation of TAAs. The authors should acknowledge it.
- Also, second major limitation is impact of concomitant csDMARDs (e.g., methotrexate) not clearly separated.
- I recommend the authors to include a graphical abstract or figure summarizing key correlations for clarity.
- I recommend the authors to expand discussion on clinical implications: Can TAAs be used for risk stratification or therapy monitoring? Please discuss how findings may translate into clinical practice (e.g., monitoring vascular risk).
- I think the authors should add a discussion of whether TAA measurements are cost effective.
Author Response
This study is novel in its investigation of tumour-associated antigens (TAAs) in the context of rheumatoid arthritis (RA) and their modulation by JAK inhibition (tofacitinib). It offers potentially valuable insights into RA pathophysiology and vascular comorbidities. This study is the first to correlate TAAs with vascular parameters (ccIMT, cfPWV) in RA. Also, this study provides longitudinal data with robust biomarker analysis, and addresses both inflammatory and vascular components of RA.
We thank the reviewer for this positive opinion agreeing that our work was worthwhile. We address all issues and marked in RED.
However, several issues need to be addressed.
- I think the major limitation of this study is no functional mechanistic studies to explain differential regulation of TAAs. The authors should acknowledge it.
As also discussed in the paper and here later, this area of TAAs has not been well studied. We know some molecular mechanisms of these TAAs, but the regulation, expression in tissues and influence of antirheumatic therapies are largely unknown. Yet, we acknowledge this as a limitation.
- Also, second major limitation is impact of concomitant csDMARDs (e.g., methotrexate) not clearly separated.
MTX or leflunomide was combined with tofacitinib in this study, which resulted many previous publications. This was the situation, which could not be changed. JAK inhibitors could be applied as monotherapy or in combination with csDMARDs, but this is also how treatment occurs in real life. What is important is effective therapy. Yet, this limitation is also added.
- I recommend the authors to include a graphical abstract or figure summarizing key correlations for clarity.
It is very difficult to show correlations on a figure so we added a simplified table showing the correlations. This Table 5 is also cited in the text.
- I recommend the authors to expand discussion on clinical implications: Can TAAs be used for risk stratification or therapy monitoring? Please discuss how findings may translate into clinical practice (e.g., monitoring vascular risk).
We now added a few sentences on possible clinical applications.
- I think the authors should add a discussion of whether TAA measurements are cost effective.
We are sorry but no cost-effectiveness analysis has been performed. In Hungary, assessment of TAAs is reimbursed in the healthcare system. As TAAs reflect inflammation and vascular pathology, it might be cost-effective to use them instead of multiple markers.
We thank the reviewer again for the very helpful comments and issues. We hope we could address all issues satisfactorily.
Reviewer 2 Report
Comments and Suggestions for Authors
The manuscript entitled “Effects of tofacitinib therapy on circulating tumour-associated antigens and their relationship with clinical, laboratory and vascular parameters in rheumatoid arthritis” reported that determined the effects of one-year tofacitinib treatment of TAA levels, as well as correlations of TAA levels with various RA-associated and vascular parameters. JAK inhibition might decrease the levels of some TAAs and increase those of others. TAA levels might be associated with RA-related and vascular biomarkers. These results suggest that TAAs might play a role in inflammatory processes and vascular pathology underlying RA. In my opinion, this manuscript cannot be published in its current state until these issues are resolved:
1.The sample size is relatively small, only 26 patients completing the study. This may limit the generalizability of the results. It is advisable to supplement the rationale for sample size estimation or discuss the potential impacts of a small capacity sample on the findings.
2. Although changes in TAA levels and their associations have been observed, the paper lacks in - depth exploration of the underlying molecular mechanisms by which JAK inhibition affects TAA levels. Consider adding research on relevant pathways or proposing mechanistic hypotheses.
3. The study only focuses on the tofacitinib - treated group, lacking a comparison with other anti - rheumatic treatment regimens. It is difficult to determine the uniqueness of tofacitinib's effects on TAA. Adding a control group for analysis is recommended.
4.Follow - up Duration Issue:The follow - up period of only 12 months is relatively short, and the long - term trends of TAA levels remain uncertain. Consider extending the follow - up period and tracking relevant indicators.
5. The discussion on translating the research results into clinical applications in the paper is relatively weak. It is suggested to strengthen the elaboration on the clinical potential and limitations of TAA as diagnostic, prognostic markers or therapeutic targets for RA.
6.Ensure that the numbering and cross - referencing of tables and figures throughout the text are consistent and clearly formatted. The caption of Figure 1 appears twice, please confirm and modify it.
Author Response
The manuscript entitled “Effects of tofacitinib therapy on circulating tumour-associated antigens and their relationship with clinical, laboratory and vascular parameters in rheumatoid arthritis” reported that determined the effects of one-year tofacitinib treatment of TAA levels, as well as correlations of TAA levels with various RA-associated and vascular parameters. JAK inhibition might decrease the levels of some TAAs and increase those of others. TAA levels might be associated with RA-related and vascular biomarkers. These results suggest that TAAs might play a role in inflammatory processes and vascular pathology underlying RA.
We thank the reviewer for this positive opinion agreeing that our work was worthwhile. We address all issues and marked in RED.
In my opinion, this manuscript cannot be published in its current state until these issues are resolved:
1.The sample size is relatively small, only 26 patients completing the study. This may limit the generalizability of the results. It is advisable to supplement the rationale for sample size estimation or discuss the potential impacts of a small capacity sample on the findings.
This study was performed 3-4 years ago (2021-2023) on 26 subjects and until now many publications have become available. When designing the study, we calculated the sample size as 30 in total. Our other analyses and publications on CV pathophysiology (doi: 10.3389/fmed.2022.1011734. ), bone metabolism (doi: 10.1007/s00198-021-05871-0.), PET-CT (doi: 10.1136/rmdopen-2021-001804.), and metabolic biomarkers (doi: 10.3390/biom12101483.; doi: 10.3389/fmed.2022.1011734.; doi: 10.1093/rheumatology/kead502.) confirmed that the sample size was OK. In this study, even 26 patients yielded to significant results. Therefore, in accordance with our previous calculations, we think that the sample size was OK.
- Although changes in TAA levels and their associations have been observed, the paper lacks in - depth exploration of the underlying molecular mechanisms by which JAK inhibition affects TAA levels. Consider adding research on relevant pathways or proposing mechanistic hypotheses.
A similar issue was also raised by Reviewer 1. Again, as also discussed in the paper and here later, this area of TAAs has not been well studied. We know some molecular mechanisms of these TAAs, but the regulation, expression in tissues and influence of antirheumatic therapies are largely unknown. Yet, we acknowledge this as a limitation and added to the limitations. We cannot further elaborate in the paper due to lack of information in the literature.
- The study only focuses on the tofacitinib - treated group, lacking a comparison with other anti - rheumatic treatment regimens. It is difficult to determine the uniqueness of tofacitinib's effects on TAA. Adding a control group for analysis is recommended.
As explained above this study was carried out in 2021-2023, the study was closed then and no control group was used. This was simply a self-controlled study when changes were compared with baseline. We had 7 publications coming from this study. Moreover, there have been absolutely no studies on the effects of biologics or other JAK inhibitors on these TAAs, so we are sorry, but at present we cannot perform a comparative analysis and discussion.
4.Follow - up Duration Issue:The follow - up period of only 12 months is relatively short, and the long - term trends of TAA levels remain uncertain. Consider extending the follow - up period and tracking relevant indicators.
Many thanks for the great idea. However the study was stopped after one year. Some of the patients might still receive tofacitinib, so a follow-up can be performed in those patients. We will try to do this in the close future.
- The discussion on translating the research results into clinical applications in the paper is relatively weak. It is suggested to strengthen the elaboration on the clinical potential and limitations of TAA as diagnostic, prognostic markers or therapeutic targets for RA.
This point was also raised by Reviewer 1. We now added a few sentences on possible clinical applications.
6.Ensure that the numbering and cross - referencing of tables and figures throughout the text are consistent and clearly formatted. The caption of Figure 1 appears twice, please confirm and modify it.
Many thanks, we now clarified and corrected this.
We thank the reviewer again for the very helpful comments and issues. We hope we could address all issues satisfactorily.
Round 2
Reviewer 1 Report
Comments and Suggestions for Authors
The authorus responded accordingly to the comments.